# A First Attempt to Produce Proteins from Insects by Means of a Circular Economy

**DOI:** 10.3390/ani9050278

**Published:** 2019-05-24

**Authors:** Silvia Cappellozza, Maria Giovanna Leonardi, Sara Savoldelli, Domenico Carminati, Anna Rizzolo, Giovanna Cortellino, Genciana Terova, Enzo Moretto, Andrea Badaile, Giuseppe Concheri, Alessio Saviane, Daniele Bruno, Marco Bonelli, Silvia Caccia, Morena Casartelli, Gianluca Tettamanti

**Affiliations:** 1Consiglio per la Ricerca in Agricoltura e l’Analisi dell’Economia Agraria, Centro di Ricerca Agricoltura e Ambiente (CREA-AA), 35143 Padua, Italy; alessio.saviane@crea.gov.it; 2Dipartimento Scienze per gli Alimenti, la Nutrizione e l’Ambiente, Università degli Studi di Milano, 20133 Milan, Italy; mgiovanna.leonardi@unimi.it (M.G.L.); sara.savoldelli@unimi.it (S.S.); 3Consiglio per la Ricerca in Agricoltura e l’Analisi dell’Economia Agraria, Centro di Ricerca Zootecnia e Acquacoltura (CREA-ZA), 26900 Lodi, Italy; domenico.carminati@crea.gov.it; 4Consiglio per la Ricerca in Agricoltura e l’Analisi dell’Economia Agraria, Centro di Ricerca Ingegneria e Trasformazioni Agroalimentari (CREA-IT), 20133 Milan, Italy; anna.rizzolo@crea.gov.it (A.R.); giovanna.cortellino@crea.gov.it (G.C.); 5Dipartimento di Biotecnologie e Scienze della Vita, Università dell’Insubria, 21100 Varese, Italy; genciana.terova@uninsubria.it (G.T.); d.bruno1@uninsubria.it (D.B.); gianluca.tettamanti@uninsubria.it (G.T.); 6Museo Vivente degli Insetti “Esapolis”— Butterfly Arc, 35143 Padua, Italy; enzo@micromegamondo.com (E.M.); andrea.badaile@gmail.com (A.B.); 7Dipartimento di Agronomia Animali Alimenti Risorse Naturali e Ambiente (DAFNAE), Università degli Studi di Padova, 35020 Legnaro (Pd), Italy; giuseppe.concheri@unipd.it; 8Dipartimento di Bioscienze, Università degli Studi di Milano, 20133 Milan, Italy; marco.bonelli@unimi.it; 9Dipartimento di Agraria, Università degli Studi di Napoli, Federico II, 80055 Naples, Italy; silvia.caccia@unina.it

**Keywords:** waste reduction index, fat body, midgut, insect meal, rainbow trout, aquaculture, microbiological analyses, drying process, earthworms

## Abstract

**Simple Summary:**

Protein production for animal husbandry is a crucial ecological problem because of its impact on the environment, as it requires water, energy, and land. These resources are limited and not reusable. In this study, we obtained a continuously regenerating system in which by-products of a process constituted rough material for another one. Leftovers from fruit and vegetable markets were employed as rearing substrate for insects (Black Soldier Fly). Insect biomass was transformed into meal and oil for fish feeding and food/pharmaceutical industry, respectively. The residuals from insect rearing were then used as substrate to grow earthworms, which transformed this material into compost for plants. Therefore, we returned to the starting point of our economic and ecological closed loop, i.e., to soil improvers (nutrient material) for fruit and vegetable production. Moreover, earthworms can be conveniently employed as fishing bites. We also studied a series of physiological parameters of the living organisms involved in this system to verify their health conditions (insects), and growth performances (insects and fish). Microbiological analyses of insects, rearing substrate, and insect meal were conducted to assess their safety for fish and humans. Related technological processes, such as insects drying, grinding, and oil extraction, were also tested.

**Abstract:**

The worldwide growing consumption of proteins to feed humans and animals has drawn a considerable amount of attention to insect rearing. Insects reared on organic wastes and used as feed for monogastric animals can reduce the environmental impact and increase the sustainability of meat/fish production. In this study, we designed an environmentally closed loop for food supply in which fruit and vegetable waste from markets became rearing substrate for *Hermetia illucens* (BSF— black soldier fly). A vegetable and fruit-based substrate was compared to a standard diet for Diptera in terms of larval growth, waste reduction index, and overall substrate degradation. Morphological analysis of insect organs was carried out to obtain indications about insect health. Processing steps such as drying and oil extraction from BSF were investigated. Nutritional and microbiological analyses confirmed the good quality of insects and meal. The meal was then used to produce fish feed and its suitability to this purpose was assessed using trout. Earthworms were grown on leftovers of BSF rearing in comparison to a standard substrate. Chemical analyses of vermicompost were performed. The present research demonstrates that insects can be used to reduce organic waste, increasing at the same time the sustainability of aquaculture and creating interesting by-products through the linked bio-system establishment.

## 1. Introduction

The worldwide growing consumption of proteins for human and animal nutrition has drawn a considerable amount of attention to insect rearing; sector stakeholders focus on insect potential to convert organic material of low quality into high quality food and feed. Using insects, and in particular black soldier fly (BSF), *Hermetia illucens*, larvae to transform waste into feed cannot be considered as an original idea. In their exhaustive review, Wang and Shelomi [1] clearly stated that the practice of rearing insects on human-inedible wastes and using these Arthropods as a protein source for domestic animals is more environmentally friendly than using the land, water, and energy resources that could otherwise be employed to produce food for humans [2]. However, to avoid possible conflicts with the EU legislation and FDA (Food and Drug Administration) in the USA, potential biohazardous wastes such as animal manure and household waste cannot be used as substrates for insect rearing. The main substrates currently used in the European insect production include former foodstuffs not containing meat and fish, and co-products from the primary production of food of non-animal origin [3]. Among the substrates categorized by the European Food Safety Authority (EFSA), in addition to animal feed materials included in the EU catalogue [4], food waste of non-animal origin from food for human consumption was also considered. In particular, by-products of crops or leftovers of fruit and vegetable discarded from markets might represent a good substrate for rearing insects, which could be then used to feed domestic animals for human consumption. The use of food/waste by-products is not only a means to generate proteins for animals and man. Food waste is an issue of global importance directly linked to environmental, economic, and social impact. Different studies showed that between 1/3 and 1/2 of the world food production is wasted [5,6]. The European Parliament adopted the resolution of 19th January 2012 [7] to point out the necessity of avoiding food wastage. Sectors that contribute to food waste are: primary production, processing, wholesale and logistics combined with retail and markets, food service, and household. The purpose of the present research was to demonstrate that it is possible to properly reduce food waste generated from fruit and vegetable markets by transforming this organic material into products that enter a circular value chain. According to the European Commission’s Circular Economy Package of 2015 “the transition to a more circular economy, where the value of products, materials and resources is maintained in the economy for as long as possible, and the generation of waste minimized, is an essential contribution to the EU’s efforts to develop a sustainable, low carbon, resource efficient, and competitive economy” [8]. 

In accordance with this statement, we designed a continuously regenerating system in which by-products of each process constituted the rough material for another one. The cycle began with fruit and vegetable market leftovers that were employed to rear BSF larvae. Afterwards, insect biomass underwent drying, defatting, and grinding to obtain oil and meal and the latter was used for fish feed formulation. Feed containing BSF meal was then tested in a 3-month feeding trial using rainbow trout (*Oncorhynchus mykiss*). Leftovers of BSF rearing cycle were subsequently employed to grow earthworms (*Eisenia fetida*), which transformed their growing litter into soil amendment for plants. Earthworms were used as bites for fishing.

In the present study we considered not only issues that are fundamental in a circular economy system, i.e., waste reduction, exploitation of different by-products, recycling of materials, but we evaluated even aspects that are important to make this closed-loop feasible, reliable and safe, i.e., insect health, insect meal acceptance by fish, nutritional composition of insect meal, microbiological quality of rearing products, and processing yields. In particular, studies on insect microbiology and health (the latter evaluated through microscopic analysis of the insect tissues and organs) were included in this paper according to the recommendations recently issued by IPIFF (International Platform of Insects for Food and Feed), which focused on good hygiene practices for the complete insect production chain and insect health [9]. Indeed, as stated in the IPIFF Guide: “Insects intended for food and/or feed have the legal status of *farmed animal*.” The general requirements of animal health, therefore, also apply to insects” [9]. Moreover, some aspects, such as the growth of BSF larvae on fruit and vegetable market leftovers and their drying were first studied at a laboratory scale and then the know-how was used to accomplish mass rearing after standardization of the method.

All the aforementioned activities contributed to building a circular activity exploiting the capability of an insect to grow on organic waste. The present research improves the overall vision of the field and provides a platform of knowledge that can be adopted for further industrial applications.

## 2. Materials and Methods

### 2.1. Insect Rearing under Laboratory Conditions

Eggs were obtained from a colony of BSF collected from the field (Lombardy, Northern Italy) in 2015, which since then has been maintained under standard conditions [10] at the University of Milan.

Egg clutches were directly positioned on the two experimental diets: (1) Standard diet (SD) (Gainesville diet) [11]; or (2) a diet composed of a mix of fruit and vegetables (zucchini, apple, potato, green beans, carrot, pepper, orange, celery, kiwi, plum, eggplant) cut into small pieces of about 5 mm (Vegetable Mix Diet, VMD); both substrates were put in plastic containers (500 mL) where a perforated bottom assured drainage of excessive humidity. Eggs were collected within 24 h from the oviposition to obtain homogenous hatching. Eggs and larvae were maintained at 27 ± 1 °C, RH 50 ± 0.5% under a 16:8 (L:D) photoperiod. A series of preliminary tests was carried out to establish the optimum quantity of feed under the aforementioned rearing conditions. Then, three replicates of 200 larvae were reared on both substrates (200 mg larva^−1^ day^−1^) (wet weight). Three samples of 10 larvae per each replicate were weighed every two days until 40% of the larvae reached the pre-pupal stage, which is indicated by the change in their color from creamy to black. Every two days, leftovers of insect feeding activity were weighed, after their removal from each container.

### 2.2. Insect Mass Rearing

Mass rearing was performed in order to produce sufficient insect biomass for fish feeding trials. The same above described laboratory conditions were maintained.

Three replicates of ≈5000 eggs were directly put on VMD diet in plastic containers (40 × 60 × 5 cm) where a perforated bottom assured drainage of excessive humidity. A series of preliminary tests was carried out to establish the optimum quantity of feed with this sample size, and it was found to be 120 mg VDM larva^−1^ day^−1^ (wet weight); starting from the tenth day after hatching, three samples of 10 larvae from each replicate were collected and weighed. At the end of the rearing cycle, BSF and rearing leftovers were removed from each container and weighed. While under laboratory conditions BSF pre-pupae were easily isolated and discarded from the rearing substrate, under mass rearing conditions stage selection was not just as simple; therefore, when BSF biomass is mentioned in the text, it refers to a mix of mostly larvae and occasionally pre-pupae.

The waste reduction index (WRI) and the overall degradation (D) for the whole experiment were calculated according to Diener et al. [12].

Three hundred replicates of ≈5.000 eggs were used to obtain BSF biomass for fish feed production.

### 2.3. Light Microscopy

Larvae at the last instar reared on SD and VMD were anesthetized on ice before dissection. Midgut and fat body were collected and immediately fixed in 4% glutaraldehyde (in 0.1 M Na-cacodylate buffer, pH 7.4) overnight at 4 °C and separately processed as follows. Midgut samples were processed according to Bonelli et al. [13]. Briefly, after postfixation in 2% osmium tetroxide for 2 h at room temperature, specimens were dehydrated in ascending ethanol series and then embedded in resin (Epon/Araldite 812 mixture). Fat body samples, after fixation, were postfixed in 2% osmium tetroxide for 1 h at room temperature, then dehydrated, and embedded in resin according to Pimentel et al. [14]. Semithin sections were obtained as described in Franzetti et al. [15] and observed with a Nikon Eclipse Ni-U microscope equipped with a DS-5 M-L1 digital camera system.

### 2.4. Drying Process

To evaluate drying procedure under laboratory conditions, the following types of frozen BSF were employed: Biomass reared on SD (600 g), BSF reared on VMD together with the leftovers of the rearing substrate (VMD1, 3 kg) and BSF reared on VMD without the leftovers of the rearing substrate (VMD2, 3 kg).

Before each drying experiment, frozen BSF were thawed at 4 °C for 18 h. A pilot alternate upward-downward air circulated drier (Thermolab, Codogno, Italy) equipped with three grids and a 3 kg maximum loading capacity was used. The drying conditions were: temperature, 70 °C (dry bulb); air speed, 8 h at 1.5 m/s, then overnight at 1 m/s; time, until constant weight. Weights were recorded using a high precision platform of 25 kg capacity (Gibertini model PTF25D, Gibertini Elettronica, Italy). Before drying, each BSF replicate was packed into a tulle bag, to avoid possible losses of biological material, due to the air flow.

For each replicate the percent residue weight (*Wr*) was computed according to:(1)$$Wr_{i}\left(\%\right)=100-\frac{W_{0}-W_{i}}{W_{0}}\times 100,$$
where *W*_0_ is the weight at time 0 and *W_i_* is the weight at the drying time (*i*).

The trends over drying time of *Wr* were studied using regression analysis (Statgraphics version 5.1 Plus, Manugistic Inc., Rockville, MD, USA) and the model with the highest performance was considered for all the experimental lots (SD, VMD1, VMD2).

The above-described drying experiments provided the set-up of optimum conditions for the mass drying of BSF coming from their mass rearing. BSF mass drying was performed at CREA-AA, Padua, using a silkworm cocoon drier (Officine specializzate Garbujo, Treviso).

### 2.5. Mass Grinding and Defatting

The BSF dried biomass obtained from CREA-AA was pressed for obtaining oil and partly defatted meal. A continuous screw press (MIG srl, Fornovo San Giovanni, BG, Italy) mod. PC 25 was used. The equipment was constituted of a loading hopper, which was filled with the dried biomass, and a conveyor belt for material transfer into the continuous heater. The biomass arrived into the heater installed over the press where the temperature was kept at 80 °C by means of indirect heating. The pre-heated material was finally transferred into the press by simple drop assisted by a vibrating device. The dried BSF biomass was pressed using a screw-press rotating at approximately 20 revolutions per minute (rpm).

To remove all solid impurities the obtained oily material was finally filtered while it was still hot.

### 2.6. Microbiological Analyses

Samples (10 g) were homogenized in sterile peptone water (90 mL, peptone 1 g L^−1^, Thermo Scientific Oxoid, Basingstoke, UK) using a Stomacher 400 apparatus (VWR International PBI, Milan, Italy), and 10-fold diluted for microbial enumeration of total mesophilic aerobes, *Enterobacteriaceae*, *Escherichia coli*, yeasts and molds, enterococci, and sulfite-reducing Clostridia spores in appropriate growth media, briefly as follows: total mesophilic aerobes in accordance with the ISO 4833-1:2013 standard method [16]; *Enterobacteriaceae* in accordance with the ISO 21528-2:2004 [17]; *E. coli* in accordance with the ISO 16649-2:2010 [18]; yeasts and molds in accordance with the ISO 21527-1:2008 [19]; enterococci were counted as reported by Mucchetti et al. [20]; sulfite-reducing Clostridia spores in accordance with ISO 7937:2004 [21], by a preliminary treatment of diluted samples at 80 °C for 10 min. The presence of *Salmonella* spp. was assessed in accordance with the ISO 6579-1:2017 [22] standard method.

All microbiological analyses were performed in duplicate. The results of the microbial counts were expressed as means of log colony-forming units (cfu) per gram of sample ± standard deviations.

### 2.7. Experimental Diets and Fish Feeding Trial

Four diets were formulated with increasing percentages of BSF meal in substitution of fishmeal (FM). The BSF meal used in the trial derived from the aforementioned insect mass rearing and mass drying. Specifically, one control diet with 0% (BSF 0) and three experimental diets with 10% (BSF10), 20% (BSF 20), and 30% (BSF30) of BSF meal, were formulated to be isonitrogenous [crude protein: about 49 g/100 g dry matter (d.m.)], isolipidic [ether extract: approximately 18 g/100 g d.m.] and isoenergetic [gross energy: about 22.90 MJ/kg d.m.]. In order to maintain diets isonitrogenous, isolipidic and isoenergetic, and due to the different chemical composition of BSF meal compared to FM, with the increase of BSF meal inclusion, soybean oil and wheat bran amounts were correspondingly modified. Feeds were prepared at the Department of Agricultural, Forest and Food Science (DISAFA), University of Turin (Italy). The apparent digestibility coefficients of dry matter, proteins, ether extract and gross energy of each diet were assessed through an in vivo digestibility trial. For more details on experimental diets and fish feeding, refer to Terova et al. [23]. A 3-month feeding trial was then performed on rainbow trout *(O. mykiss)* fed twice daily with the four experimental diets. Feed was manually distributed and feeding rate was restricted to 1.5% of biomass for the entire duration of the experiment. At the end of the feeding trial all fish were individually weighed to calculate weight gain (WG = final body weight − initial body weight).

### 2.8. Nutrient and Mineral Analysis

Nutrient analysis was performed on BSF meal obtained from mass-reared insects according to the methods of sampling and analysis for the official control of feed [24]. The La. Chi.’s protocol (DAFNAE, University of Padua) was adopted for starch and fatty acids. In particular, for starch the method was based on the enzymatic digestion of starch and following determination of glucose by HPLC. The chitin content was analyzed according to Finke [25]. The nitrogen content of the chitin fraction was determined, and protein values corrected [26]. For fatty acids, fat extraction from the sample was done according to the Folch et al. method [27]. The fat extract was analyzed after extraction with diethyl ether with the Soxhlet system [26]. Amino acids were calculated according to standard methods [27,28,29].

A mineral analysis was performed on BSF meal obtained from mass-reared insects and experimental fish diets, following the method described by AOAC [30]. The samples were dried to constant weight in an oven at 70 °C and then each sample was divided into three equal parts and mineralized in a microwave oven (Ethos one-Milestone) by adding HNO_3_ at 70%, and H_2_O_2_ at 30% (Sigma Aldrich, Italy) to each part of the samples. The resulting solutions were transferred into 25 mL flasks making up the volume with milliQ water (Merck Millipore). All reagents were metal trace analysis grade. Analyses were performed on a Thermo Scientific atomic absorption spectrometer: three replicates of each sample gave RSDs ≤ 5%.

Ca, Cu, Fe, Mg, Mn, and Zn were determined at mg/g dry matter levels by flame atomic absorption spectrometry (F-AAS) with deuterium lamp background correction.

As, Cd, Co, Cr, Ni, and Pb were determined at µg/g dry matter levels by graphite furnace atomic absorption spectrometry (GF-AAS) coupled with Zeeman background correction.

Blank samples were added to each analysis and the instrumental detection limits were determined as well. For quantitative analysis, sets of appropriate calibration solutions were prepared from standard solutions (J.T.Baker—Fisher Scientific, Milan, Italy) of each element. All the calibration curves gave correlation coefficients (R^2^) ≥ 0.99.

### 2.9. Earthworm Rearing

Three rearing substrates (experimental theses) to raise earthworms were prepared and put into separate containers. The test was carried out in three replicates; therefore, nine plastic boxes were prepared. The first substrate was composed of the dried leftovers of BSF rearing (250 g; 90% d.m.) and 750 g of peat moss; the second was prepared by mixing 750 g of the same fruit and vegetable material used as rearing substrate for BSF larvae (20% d.m.) and 750 g of peat moss; the fresh material was minced and let one month in the container to obtain a first partial decomposition; the third substrate, which was used as control, was composed of peat moss only.

*E. fetida* specimens were collected from the field in the Veneto region (Northern Italy); they were chosen approximately of the same size, washed, wiped and weighed, so that each replicate of the three theses was composed of ten earthworms of similar weight. When earthworms were placed in the plastic containers, 500 mL of water were also added to each replicate; afterwards, water was added in the same quantity for all the containers at regular intervals, when the loss of humidity was detectable at touching. The environmental temperature in the room was maintained at 28 ± 1 °C. After 45 days the earthworms were separated from their feeding substrates, counted, and weighed. Their substrates were chemically analyzed to establish their value as soil improvers (see Section 2.10).

### 2.10. Substrate Analysis before and after Earthworm Rearing

The analysis was performed according to ANPA [31]. Compost C, N and S content were measured using a CNS automatic analyzer (Elementar vario MACRO CNS, Elementar Analysen systeme GmbH, Hanau, Germany). Organic-N was calculated by subtracting NH^+^_4_ (determined by the selective electrode, Sevenmulti Mettler Toledo) from total Kjeldahl N [32].

### 2.11. Statistical Analysis

The following software programs were used for statistical analyses: Statistica 8 (StatSoft), IBM SPSS Statistics version 25, GraphPad Prism5 (GraphPad Software, Inc., La Jolla, CA, USA). One-way ANOVA, followed by Tukey’s test, was carried out. Legends to figures and tables give details about statistical methods.

Chemical, nutritional, and microbiological analyses were performed in duplicate.

## 3. Results

### 3.1. Insect Rearing

Under laboratory conditions, the comparison between the development of BSF larvae reared on SD or VMD showed that the mix of fruit and vegetables represented a less nutritious substrate than a more balanced SD. In fact, although the maximum weight achieved before the prepupal stage was not significantly different in the two groups, the duration of the larval stage was longer in VMD group than in SD group (Figure 1). More precisely, SD larvae needed to feed for about 17 days to reach the prepupal stage with a final individual weight of 225.4 ± 7.3 mg, while the time interval was more than doubled for the VMD group, with a final individual weight of 183.3 ± 6.6 mg. Nevertheless, when VMD larvae were mass-reared, their growth performances ameliorated, as shown in Figure 2. The time necessary to reach the prepupal stage diminished in comparison to laboratory conditions (31 vs. 45 days, respectively) and the final weight was 230.8 ± 26.9 mg. Since BSF larvae have a gregarious behavior, the improvement in larval growth performance in mass rearing condition is likely related to the crowded environment (120 mg VDM larva^−1^ day^−1^ in mass rearing condition vs. 200 mg VDM larva^−1^ day^−1^ in laboratory rearing condition); in fact, the definition of the optimum quantity of feed under mass rearing conditions was the result of a long series of tests (data not shown), which were carried out to minimize feed waste and rearing leftover management and to maximize insect growth performance.

WRI and D for the whole experiment were calculated. It is noteworthy that D, in the mass rearing on VMD, reached 96% although the WRI was rather low (Table 1). However, the reason why WRI resulted low entirely depends on the fact that this index is the ratio between D and the days spent by the larvae to reach the pre-pupal stage. As BSF larvae delayed their larval development when reared on VMD, even under mass rearing, the obvious result was a decrease in WRI, although D was very high.

### 3.2. Insect Welfare and Health Status

To evaluate the health status of the larvae grown on VMD and the suitability of a fruit and vegetable mix as feeding substrate in mass rearing, we performed morphological analyses on the midgut and fat body, two organs that are involved in food digestion, nutrient absorption, nutrient storage, and immune response. Tissues isolated from larvae grown on VMD were compared to those taken from larvae reared on SD, whose composition guarantees an optimal development of BSF larvae. Both organs were subjected to microscopy analysis to assess whether the feeding substrate could affect their morphology. The posterior midgut of larvae reared on VMD showed differences compared to larvae grown on SD. In particular, columnar cells (Figure 3B) were characterized by a brush border that was thicker than that observed in larvae reared on SD (Figure 3A). Moreover, by comparing the fat body, differences in tissue morphology were observed. In fact, trophocytes of larvae reared on SD (Figure 3C) showed a higher number of protein granules, but a lower dimension of lipid droplets, compared to larvae grown on VMD (Figure 3D). This evidence confirmed the data reported in other works [14,33].

### 3.3. Insect Processing

#### 3.3.1. Set-Up of Drying Conditions at a Laboratory Scale

At the end of the drying process Wr values were 35.7 ± 0.1% for SD BSF after 360 min, and 26.1 ± 0.6% and 28.6 ± 0.2% for VMD1 and VMD2 BSF after 420 min, respectively. After the overnight drying at the lowest airflow rate, Wr values were 25.4 ± 0.1% for VMD1 BSF and 28.2 ± 0.2% for VMD2 BSF. At the end of the drying process, insects in the BSF biomass of SD e VMD2 were well separated one from each other, while VMD1 insects, due to the presence of some rearing substrate, resulted in a conglomerate difficult to crumble (Figure 4).

For all the three theses data were modelled considering for VMD1-2 the drying times up to 420 min; the model which best fitted the drying process was the Square root-X one (Y = a + b × sqrt(X)). Table 2 reports the results of the regression analysis

During the first 2 h of drying process, probably because of the different loadings of drier, Wr linearly decreased in SD biomass (R^2^_adj_ = 97.6%), while for VMD1 and VMD2 biomass (maximum loading capacity) the decrease of Wr over time followed the Square root-X model (VMD1, R^2^_adj_ = 97.6%; VMD2, R^2^_adj_ = 96.6%).

Figure 5 shows the fitted model, the confidence limits at the 95% probability level, the prediction limits and the experimental data for each experimental group.

The laboratory experiments were also designed to evaluate the effect of the rearing substrates on the drying efficiency of BSF biomass. Drying profiles of SD, VMD1 and VMD2 insect biomass were compared at different timing. Results are reported in Table 3.

No differences among the drying behaviors of the insect biomasses were recorded until the 180th min of processing time. However, from that moment the drying process was faster for BSF biomass reared on VMD (with or without rearing residuals) than on SD. This is an important finding as it is related to the energy demand of the dryer.

#### 3.3.2. Mass Drying and Oil Extraction Yields

With the aim of producing fish feed, 217 kg of fresh BSF biomass from larvae mass-reared on VMD, were dried. The obtained dried biomass was 72 kg; therefore, the yield was 33%, which was very similar to the value (28.2%) calculated in the aforementioned laboratory experimental conditions.

This dried biomass (72 kg) was analyzed before mass grinding and defatting, to evaluate total water and oil content, which resulted in 1.90% moisture and 35.62% crude ether extract. Thus, the total biomass was processed obtaining 18.8 kg of oily material (87.0% oil, 1.7% humidity, 11.3% impurity), and 53.2 kg (73.9% of the initial dry larvae amount) of the partly defatted meal (humidity:1.96%, crude ether extract:17.45%). Therefore, an oil recovery of 64% was calculated.

### 3.4. Microbiological Analyses

After the first tests under laboratory conditions (data not shown), all the microbiological analyses were performed on mass-reared BSF biomass, on its leftovers, on the derived meal, and fish feeds.

The results of the microbiological analysis of the feeding substrate (VMD), dried BSF biomass, defatted BSF meal and experimental fish feed at the four levels of insect meal addition (i.e., BSF 0, BSF 10, BSF 20, BSF 30) are reported in Table 4. The total mesophilic aerobes count was the microbial parameter characterizing all the products analyzed. The highest counts were observed in the fruit and vegetable waste substrate (6.55 ± 0.57 log cfu g^−1^); they decreased of 1 and 2 log cfu g^−1^ in the dried biomass and defatted meal, respectively, and were kept at levels of about 4 log cfu g^−1^ in all the experimental fish feeds prepared (Table 4). *Enterobacteriaceae*, present at a low level in the rearing substrate (1.70 ± 0.71 log cfu g^−1^), were under the detection limit of the count method after the processing steps for the preparation of the dried biomass and insect meal. They reached levels ranging from 1.00 ± 0.00 to 2.76 ± 0.49 log cfu g^−1^ in the experimental fish feeds. *E. coli* counted at 44 °C, regarded as indicators of contamination with enterobacteria of faecal origin, was never detected (<1.00 log cfu g^−1^). *Enterococci*, another group of bacteria that could be used as indicators of faecal contamination [34], followed the same trend of *Enterobacteriaceae*. Yeasts were enumerated at low levels in the fruit and vegetable waste substrate and dried pupae (2.48 ± 0.76 and 2.30 ± 0.43 log cfu g^−1^, respectively), while mold counts were under the detection limit in almost all samples (<2.00 log cfu g^−1^) (Table 4). The spores of sulfite-reducing *Clostridia* were present in the rearing substrate (2.15 ± 0.75 log cfu g^−1^) and were able to survive during processes for insect meal preparation (1.54 ± 0.51 log cfu g^−1^). They were found in only a few samples of fish feeds at low levels. The analyses for the detection of *Salmonella* sp.pl. revealed the absence of the bacterium in 25 g of each sample.

### 3.5. Nutritional and Chemical Analysis

Nutritional and chemical information on nutrients, amino acid content and minerals are essential information to prepare fish feeds; in fact, any of the tested diets containing BSF meal should support fish growth.

Table 5 shows BSF contribution in terms of proteins, fat and sugars, although the final meal was partly defatted (still containing 17.45% of oil). Thus, the table refers to the whole insect composition, while fish meal preparation included the aforementioned defatting process. It is important to notice that a percentage of 4.02% is constituted by chitin. Protein amount is good, being 39.42% of the total dry matter (d.m.), i.e., in the range of the protein amount found by Spranghers et al. [26], which varied from 39 to 41% for different substrates, including vegetable sources. Even nitrogen-free extracts, fat and ash contents are comparable to the values reported by Spranghers et al. [26] for BSF reared on a similar substrate.

However, previous preliminary analyses (data not shown) highlighted a lower content of protein (5–6% less out of the total d.m.) and a corresponding increase in the fat amount, in the BSF biomass obtained from larvae reared on VMD, in comparison to SD. Spranghers et al. [26] also investigated the amino acid content in BSF pre-pupae, accordingly with the different rearing substrates. Glutamic acid, aspartate and arginine are the most represented amino acids in our BSF biomass reared on VMD (Table 6), while the same authors, on a similar substrate, individuated glutamic acid, aspartate, and leucine as the most relevant amino acids.

The mineral level was not measured on BSF biomass, but on the defatted meal obtained from BSF biomass (Figure 6). Mineral levels in the experimental diets are presented in Figure 7. BSF meal contained a high level of calcium followed in a decreasing way by Mg, Mn, Fe, and Zn (Figure 6). Most insect species contain a low level of Ca (less than 3% dry matter) as they have an exoskeleton primarily composed of protein and chitin [35]. However, BSF larvae represent an exception since they have a mineralized exoskeleton in which Ca and other minerals are incorporated into the cuticle [35], and our data are consistent with the Ca profile previously reported for this species [26]. In agreement with previous studies, BSF appeared to contain high levels of Mn and Zn, too, in comparison to other insect species. According to Finke and Oonincx, [36] this is related to elevated levels of Mn and Zn in their mandibles, presumably to harden these appendages to better exploit hard feeding substrates. However, wild-caught insects appear to contain significant amounts of manganese, too [36]. With regard to copper, chromium, cobalt, nickel and the heavy metals, such as cadmium and lead, their levels in BSF meal were in the range of µg/g dry matter (Figure 6).

The mineral levels of the four experimental diets prepared using BSF meal are presented in Figure 7. There were significant differences in the levels of Ca, Mn, Fe, Zn, Cr, Cd, and Co between the diets, whereas the levels of the other minerals (Mg, Cu, Ni, Pb) were comparable.

### 3.6. Fish Growth Performances

During the 12 weeks of feeding, fish promptly accepted the experimental diets and the mortality was negligible, i.e., lower than 1%. At the end of the feeding trial, all fish had tripled their initial body weight, and growth performance indexes such as weight gain (Table 7), and standard growth ratio (SGR) [23] were not affected by diet composition. Similarly, feed conversion ratio (FCR) was comparable among the treatments and remained lower than one in all groups, meaning that all fish grew efficiently, and including the *H. illucens* meal did not negatively affect diet palatability. For more details on fish growth parameters, refer to Terova et al. [23]. With regard to the nutrient bioavailability, no differences were found among treatments for the apparent digestibility coefficient of dry matter, proteins, ether extract, and gross energy [23]. Crude protein digestibility was high and above 90% in all rainbow trout fed different experimental diets.

### 3.7. Earthworm Growth Performances

Rearing earthworms on the substrate, including BSF leftovers did not produce any negative impact on the animal biomass; on the contrary the highest trend in the biomass increase over time was recorded in the BSF thesis with respect to the addition of the simple fruit and vegetable mix (Table 8). However, the number of earthworms obtained at the end of our experiment tended to be the lowest in the BSF thesis, although again difference with VMD was not significant. Earthworms left in the boxes containing only peat significantly decreased in their number and biomass, because of starvation leading part of them to death (Table 8).

Although the duration of the experiment was too limited to outline any difference regarding the reproductive behavior of earthworms according to the rearing substrate, it is quite interesting that Dominguez et al. [37,38] showed that some vermicomposting earthworm species raised on different diets modulate their growth, reproduction or both according to the food they receive. Aira et al. [39] suggested that earthworms in low C/N diet conditions invested preferentially more energy in growth than in reproduction. In fact, in the experiment described by the authors the number of immature earthworms increased in the treatment with the highest C/N ratio, while in the treatment with the lowest C/N ratio the earthworm population was mainly composed of mature earthworms. Therefore, to better understand our results we analyzed the C/N ratio of the different substrates. Table 9 shows the nutrient contents of the three original substrates (BSF leftovers, VMD, peat moss) and those of the three organic materials (mixed BSF and peat moss, mixed VMD and peat moss, only peat moss), after the growth of the earthworms.

The ratio C/N is the lowest for BSF leftovers and it also remains the lowest after the earthworm development, despite the addition of peat moss. Therefore, there might be a relationship between earthworm behavioral data and substrate composition in terms of the C/N ratio, which might be further investigated in the future. According to the UDSA natural resource conservation service [40], materials with a C/N ratio greater than 24:1, when added to the soil, will result in a temporary nitrogen deficit (immobilization), and those with a C/N ratio lower than 24:1 will result in a temporary nitrogen surplus (mineralization). Therefore, the soil amendment coming from the joint action of BSF and earthworms appears to be optimal to be used to supply organic matter to plants.

## 4. Discussion

Fruit and vegetable leftovers, which are usually discarded from wholesale markets, represent a cheap substrate available in huge quantities that can be used to rear insects and obtain insect meal to be included in feed formulation. According to the results shown in Table 1, *H. illucens* larvae can grow on this waste material and transform it quite completely, therefore, this insect can be advantageously employed to reduce food loss in this market sector. The comparison of laboratory and mass rearing tests (Figure 1 and Figure 2) demonstrated that it is possible to rear BSF larvae on a large scale on this substrate and that they actually thrive when are in crowded conditions because they are gregarious insects. The influence of rearing density on insect performances, food consumption, and digestibility is well-known. Norris [41] found that the Approximate Digestibility index was higher in crowded *Schistocerca gregaria* adults than in isolated individuals; *Hyphantria cunea* larvae reared individually or in small groups grew somewhat more slowly than larvae reared in large groups [42]; Long [43] assessed that crowded *Plusia gamma* completed their larval stage quicker than isolated individuals. However, the underlying mechanism of the better growth of crowded BSF larvae might also be related to the regulation exerted by temperature on larval intestinal enzymes. In fact, according to Bonelli et al. [13] the optimum temperature for proteolytic activity in the midgut of BSF larvae is 45 °C. This temperature can be generated by larval overcrowding and heat produced by their movement [44]. As this feature strongly contributes to the efficiency of feeding substrate bioconversion by BSF larvae, a suboptimal density, such as that occurring under laboratory conditions, can impair the digestion processes in BSF larvae and result in a delayed growth compared to mass rearing. Notwithstanding we proved that BSF were able to transform the fruit and vegetable mix, there are many hints that the substrate is of primary importance for insect development in terms of larval cycle length, larval size and weight, and larval nutritional composition [11,45,46,47,48]. Therefore, in addition to analysing growth performances of larvae on different feeding substrates (VMD compared to SD), we focused on some physiological markers associated with the different feeding conditions. Even though our data demonstrated that VMD is suitable for rearing BSF larvae, a lower developmental rate was observed compared to SD. The lower nutritional content of VMD must be considered among the factors that can contribute to this difference. Nevertheless, the larvae are able to exploit this substrate thanks to some physiological and morphological modification of their midgut [33], in particular the length of microvilli present in the apical membrane of cells in the posterior region of the midgut (Figure 3A,B). This region is specifically involved in nutrient absorption [13] and the increase in the length of microvilli can ensure a higher absorbing surface to maximize the intake of nutrient even when the diet has a low nutritional content. Although BSF demonstrates the capacity to adapt to the food substrate to some extent, it is important to assess the long-term effect of the use of this diet on insect growth. Deficiencies of some nutrients might accumulate across generations and epigenetic effects can also become detrimental after a few reproduction cycles. In this case, a possible solution to the problem might be to preserve a genetic stock for egg reproduction through rearing on SD and to use the VMD diet only to produce insect biomass for commercial purposes.

The insect microbiota and its safety for insect transformation into feed are influenced by the nature and the hygienic conditions of the substrate and the farming environment [2,49,50]. Given these considerations, one of the aims of our work was to assess the microbial quality of the product obtained from BSF biomass through the entire production chain: From the feeding substrate to the intermediate product (insect biomass), and from the insect meal to the fish feed formulated using insect meal. The vegetable waste used to rear BSF larvae showed rather higher counts for some microbial indicators such as the total mesophilic aerobes, enterococci, sulfite-reducing anaerobic spores and yeasts. However, these microbial populations were within the ranges found on fresh-cut fruits and vegetables [51]. Other specific indicators of enteric contamination such as *E. coli* and *Salmonella* spp. were below the detection limit of the culture-dependent methods used. Raw insects are known for the elevated bacterial (10^7^ cfu g^−1^ of total aerobic bacteria, 10^6^ cfu g^−1^
*Enterobacteriaceae*) and fungal (10^4^ cfu g^−1^ yeasts and molds) load, due to the presence of microorganisms on both the animal’s surface and inside their gastrointestinal tract [52,53]. Processing of insects for production of food or feed by applying heat and pressure allowed to reduce the microbial loads and safety risks [3]. The microbiological quality of dried edible insects depends on the treatment conditions applied. The time/temperatures combination during the drying process represents the most effective way to reduce most bacterial and fungal counts [54]. The process applied in the present study for the production of insect meal started with the drying of BSF biomass at a temperature of 70 °C for the time necessary to reach a steady weight. This temperature represents a compromise between an unavoidable partial loss of nutritional function by some proteins and enzymes, which might be denatured, and the need to reduce the time required by the drying process. Afterward, a treatment for oil extraction was applied to reduce possible problems related to oxidation of insect meal and to obtain a product that can be used for the cosmetics industry or, again, for animal feeding. Finally, the defatted insect meal was used as a protein ingredient for fish feed formulation. The BSF drying and the processing to obtain the defatted insect meal resulted in a reduction of the microbial load with only the total aerobic bacteria and the sulfite-reducing clostridia spores at levels of about 4 and 1 log cfu g^−1^, respectively. Other microbial indicators such as enterobacteria and fungi were no longer detected, thus demonstrating the important role of the manufacturing processes on the reduction of microbial contamination. Furthermore, the absence of *Salmonella* spp. (in 25 g) and *E. coli* (<1.0 log cfu g^−1^) at different stages of the production chain is in compliance with the safety and process hygiene criteria applied by the European Community for foodstuffs [55], to which we refer for lack of specific criteria for insects [56]. In a survey on processed edible insect products [57], Grabowski and Klein [56] found all samples free of salmonellae, probably due to the efficiency of processing. However, dried and powdered insect products, evaluated according to process hygiene criteria applied by the Netherlands safety authorities [58], displayed high counts of total aerobic bacteria, *Enterobacteriaceae*, yeasts and mold than in our study.

The processing of feed ingredients can reduce the microbiological risks entering the food chain by applying thermal and non-thermal inactivation treatment. However, another aim of the present study was to increase the knowledge of how the insect diet could impact this industrial processing, i.e., BSF biomass drying and oil extraction. Therefore, the analysis of drying profiles related to different diets and conditions (washed or non-washed BSF biomass) was performed. The first finding is related to the convenience in operating a good separation between BSF and its substrate. In case this operation is not done before processing, the final product becomes very difficult to separate (Figure 4). Moreover, drying is important in terms of energy consumption and the duration of the process is related to its economic convenience. According to Spranghers et al. [26], the content of moisture was not significantly different in BSF pre-pupae reared on substrates of various origins (including vegetable matter), being around 60% of the total fresh biomass. This is the most important parameter to consider in the drying process. In our laboratory experiments, the percent residue weight (*Wr*) of BSF biomass was approximately the same for all the theses and the differences in the drying profiles appeared to be mostly related to the initial insect mass than to the different content of water. The large-scale drying, which had a yield of 33% d.m. on the total fresh insect biomass, confirmed this finding. These data perfectly agree with the study by Tschirner and Simon [59], who found about 30% d.m. content for BSF prepupae reared on control and protein diet. Although the water content did not vary significantly in the different drying theses, and the d.m. content was in agreement with literature data, the comparison of drying profiles highlighted interesting differences. In fact, BSF biomass coming from VMD rearing was faster in drying than BSF biomass originated by SD rearing, starting from the 180th minute of the heating process. This phenomenon is particularly remarkable in a circular economy system because it involves considerations about energy and, therefore, cost saving. In Spranghers et al. [26], fat content is higher in pre-pupae reared on vegetable waste (about 37% d.m.) than in those reared on chicken feed or digestate (34 and 22% d.m., respectively). Our nutritional analysis of VMD BSF biomass assessed that oil content was 35.62%. Lipid and protein content in fat body cells showed modifications depending on the rearing substrates. Fat body of larvae reared on SD (Figure 3C) showed a higher number of protein granules, but a lower dimension of lipid droplets, compared to larvae grown on VMD (Figure 3D). Although morphological analysis of BSF organs provides an indication on the quality more than on the quantity of reserves present in the insect cells, it corroborates the hypothesis that a different quantity or distribution of the fat and protein amounts in the BSF reared on different substrates can affect the speed of the drying process. Preliminary comparative analyses (data not shown), demonstrated that BSF biomass originated from VMD has a reduced protein content and an increased fat amount in comparison to SD. These findings support the aforementioned morphological data.

After drying, another basic process is oil extraction, which has the aim to produce defatted insect meal to increase its shelf-life, especially meal oxidative stability. BSF oil is also an important economic resource, because it can represent an alternative to fish oil that is used to feed carnivorous fish or other animals, due to its high lipid content [1]. However, BSF inclusion in fish and poultry diet can negatively affect the fatty acid profiles of their meat, because the polyunsaturated fat content tends to decrease, while saturated and monounsaturated fat content tends to increase [60,61,62]. Therefore, in our study, the efficiency of oil extraction from the insect biomass was analyzed. Few reports on the extraction yield of oil obtained from insect mechanical pressing are available in the literature. However, the percentage of oil recovered (64%) in this study by mass oil extraction, was satisfying if compared to the 40% oil recovery obtained by Surrendra et al. [63]. Once the insect meal is obtained, an important aspect is the evaluation of its nutritional value for monogastric animals. In the last few years, the aquaculture industry has become increasingly interested in using insects to replace fishmeal in aquafeeds. Reviews on the use of insect meal as a feed ingredient in farmed fish are available [64,65]. Recently, some studies investigated the effects of partial fishmeal substitution with BSF meal on fish growth performances showing that insect meal effectively supports fish growth [64,65,66,67,68]. On the basis of the chemical analysis presented in the Results, four fish diets were prepared with different substitution levels of fishmeal with insect meal. In the feeding trial, all the experimental diets were well accepted (no palatability issues) by rainbow trout that tripled their initial body weight. In line with previous studies conducted on rainbow trout [67,68], dietary inclusion of defatted BSF meal up to 30% did not affect growth performance. Indeed, in all treatments, the weight gain was very good, and no differences were found among dietary fish groups. In salmon, BSF could replace up to 100% of dietary fishmeal (corresponding to 25% of inclusion) without any significant effect on growth if methionine and lysine are supplemented to the diet [65]. In yellow catfish, a diet in which fishmeal protein was replaced by 25% of BSF meal gave the greatest growth performance and immune indices compared with the control diet containing 40% of fishmeal [66]. The reduced fish growth rate is usually due to a deficiency in lysine or methionine or to an imbalanced level of essential amino acids in the diet. In our feeding trial, the amino acid profile of experimental diets correlated well with trout requirement values, showing lysine and methionine levels comparable to control fishmeal-based diet, independently of BSF meal dietary inclusion.

At the end of the insect rearing process, leftovers resulted from the feeding activity of BSF larvae, were employed for the earthworm rearing. Earthworms recycle organic matter, increase nutrient availability, and improve soil structure, thus providing a service to the ecosystem. Vermicompost is the product of the composting process carried out by the worms and can be used for organic agriculture. In the past, attempts were carried out to grow earthworms and BSF contemporaneously. However, earthworms inoculated before organic waste decomposition by BSF could not survive, probably because of the acidic environment and high temperature created by the larvae and their excreta [69]. Furthermore, it is well known that BSF larvae can modify the microflora of the substrate in which they live, as they produce bactericidal, bacteriostatic, and fungicidal compounds [70]. This could be particularly detrimental for earthworms, as there is evidence that microorganisms provide food for them; while bacteria are of minor importance in their diet and algae are of moderate importance, protozoa and fungi are major sources of nutrients, as associated to earthworms’ intestinal tract [71]. Although a joint action with BSF is not possible, earthworms can intervene when the development of BSF larvae is completed, usefully employing the residuals of larval activity for their own growth and soil amendment production, as shown in Table 8 and Table 9. Earthworm rearing was performed for 45 days only; therefore, data presented in this work are preliminary and had the aim to test the possible negative effect of BSF leftovers on earthworm development. In the tested ratio with peat moss, BSF leftovers not only did not affect earthworm survival, but also sustained their growth at least as the fruit and vegetable waste themselves. In general, the BSF leftovers are richer in N in comparison to fruit and vegetable wastes; therefore, they are positive starting material to lower the C/N ratio of the amendment. According to the Italian legislation [72], the ratio C/N in plant amendment should be <25. This is the case of our mix of peat moss and BSF leftovers, after the earthworm action. Although this is not the only parameter that must be considered to define the product of the composting action as a plant amendment, it confirms the process feasibility.

## 5. Conclusions

Research on the use of insect for feed and food is still at its very beginning. However, in the last ten years an increasing interest in using insects for both these economic sectors became tangible. Private entrepreneurs engaged themselves in this activity and many industrial insect rearing companies can nowadays produce huge quantities (tons) of insects a day. On the other hand, environmental benefits of using insects as food and feed have been mostly declared than become facts and the safety of the whole processing chain is still far from reaching a general, and shared, soundness.

This paper does not have the ambition of describing a complete series of processing actions in details for the immediate transfer to large-scale production. In fact, many large-scale production processes for insect production, do already exist. On the other hand, these processes very often fail to obtain good results in terms of human security, environment, ethics applied to animal welfare. The final goal of our research was to delineate an interdisciplinary method, which can also be applied to other insects, based on the principles of the circular economy and with the objective to save natural resources, provide feed and food for animals and humans, create new technologies and job opportunities, and, at the same time, avoid microbiological risks and any kind of hazards for human health.

## Figures and Tables

**Figure 1 animals-09-00278-f001:**
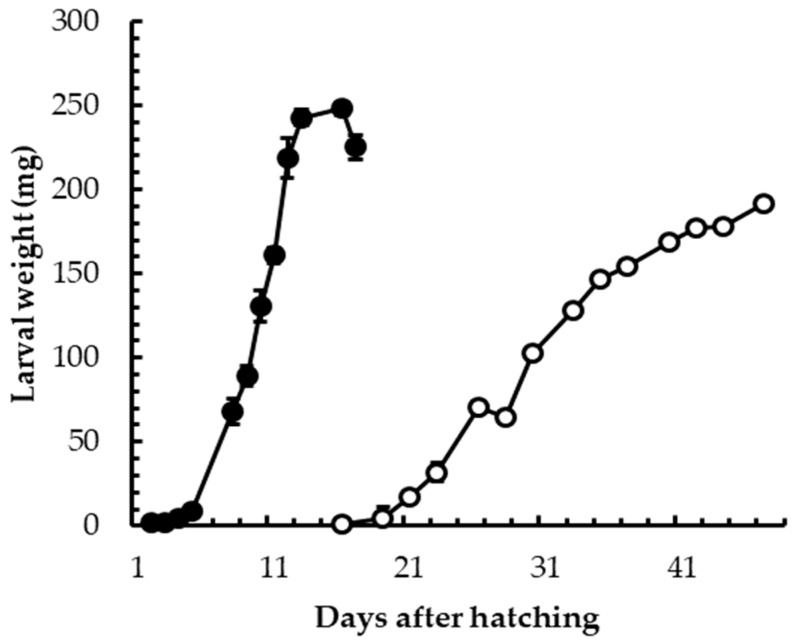
Development time of black soldier fly (BSF) larvae grown on SD (black circles) and VMD (white circles under laboratory conditions. Bars refer to standard deviation.

**Figure 2 animals-09-00278-f002:**
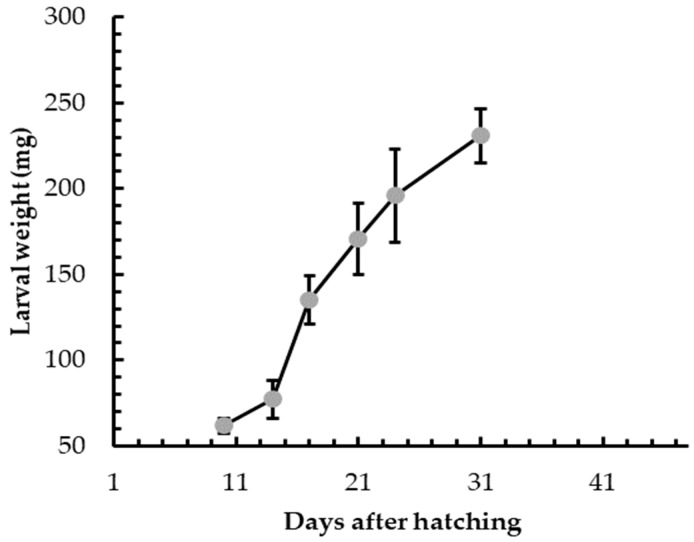
Development time of BSD larvae grown on VMD in mass rearing conditions. Bars refer to standard deviation.

**Figure 3 animals-09-00278-f003:**
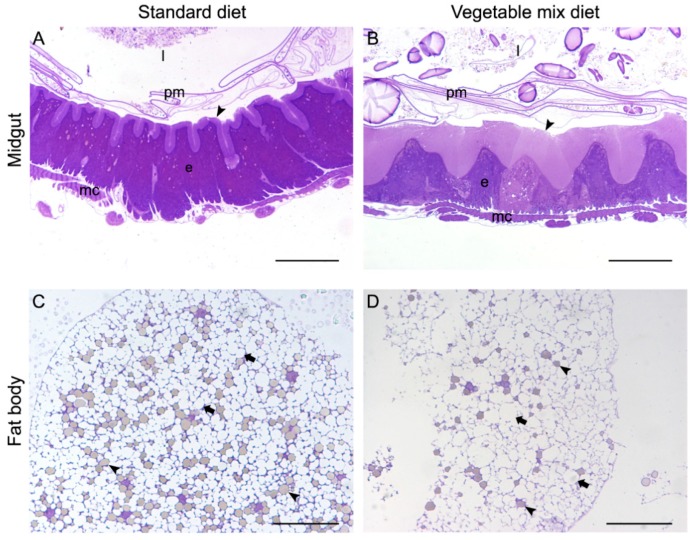
Morphological comparison of midgut and fat body from larvae reared on SD and VMD. (**A**,**B**) Cross-section of the posterior midgut. Columnar cells display a different thickness of the brush border (arrowheads). (**C**,**D**) Fat body. A higher amount of protein granules (arrowheads) and a smaller size of lipid droplets (arrows) is observable in trophocytes of larvae grown on SD compared to VMD. e, epithelium; mc, muscle cells; l, lumen; pm, peritrophic matrix. Bar: 50 μm.

**Figure 4 animals-09-00278-f004:**
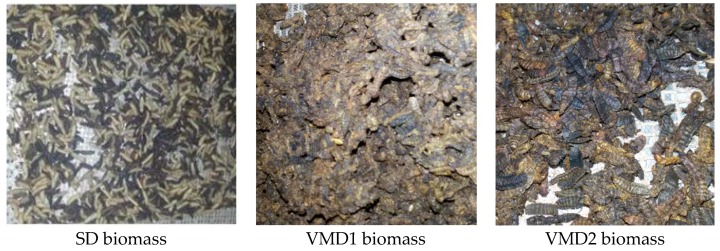
BSF biomass reared on SD and VMD and dried with (VMD1) or without (VMD2) substrate, as it appeared at the end of the drying process.

**Figure 5 animals-09-00278-f005:**
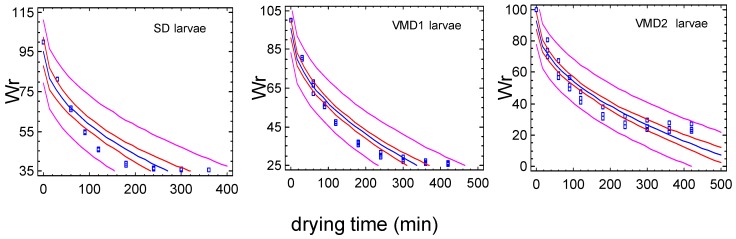
Square root-X models of percent residue weight (Wr) vs. drying time: Fitted model (blue line), confidence limits at 95% level (red lines), prediction limits (purple lines) and experimental data.

**Figure 6 animals-09-00278-f006:**
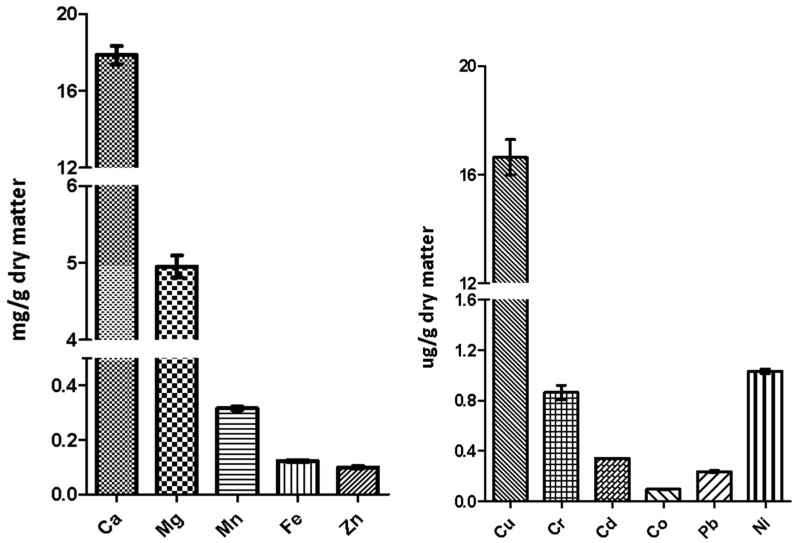
Mineral levels in BSF meal. One-way ANOVA and Tukey’s post-hoc test (GraphPad Prism5).

**Figure 7 animals-09-00278-f007:**
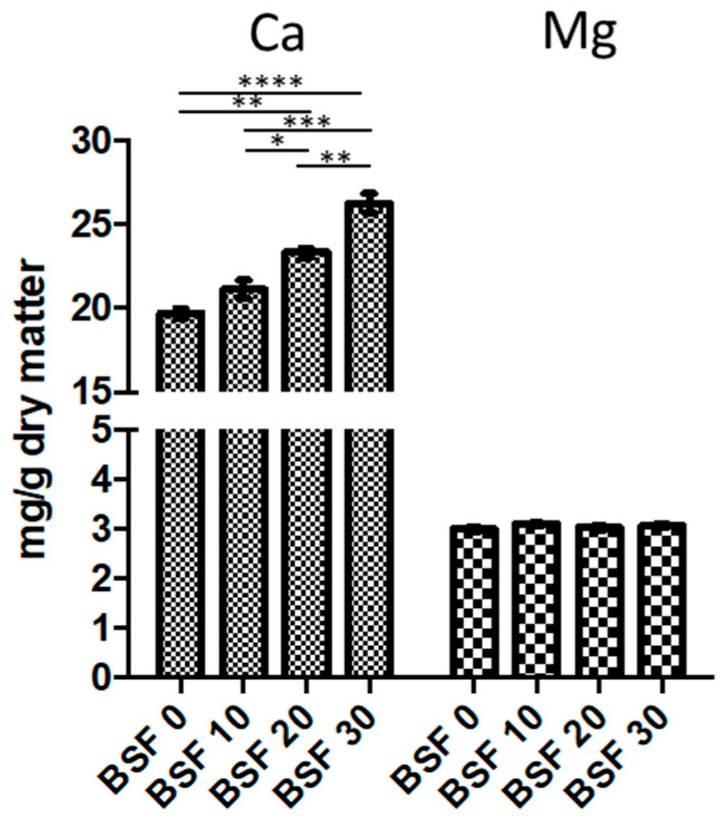
Mineral levels in the fish diets (one-way ANOVA and Tukey’s post-hoc test (GraphPad Prism5).

**Table 1 animals-09-00278-t001:** Larval developmental time and final weight, overall degradation and waste reduction index of BSF larvae grown on different diets and conditions.

	SD	VMD	Mass Rearing VMD
Development time (day)	17 ± 0 c	45 ± 1 a	31 ± 0 b
Maximum weight before prepupal stage (mg)	225.4 ± 7.3	183.3 ± 6.6	230.8 ± 26.9
Overall degradation (D)	0.46 ± 0.01 c	0.52 ± 0.01 b	0.96 ± 0.00 a
Waste reduction index (WRI)	3.06 ± 0.06 a	1.78 ± 0.07 c	2.35 ± 0.00 b

Values are reported as mean ± SD. In the same row different letters denote significant differences. One-way-ANOVA and Tukey’s post-hoc test were carried out by using IBM SPSS Statistics version 25.

**Table 2 animals-09-00278-t002:** Results of the regression analysis of percent residue weight (Wr) vs. drying time.

Larvae	N_obs_	*Intercept (a)*	*Slope (b)*	*r*	R^2^_adj_	SEE	MAE
		Estimates	Estimates				
SD	16	95.09 ± 3.43	−3.65 ± 0.28	−0.96	91.07	6.71	5.60
VMD1	30	95.78 ± 2.27	−3.86 ± 0.17	−0.97	94.69	5.68	4.88
VMD2	30	92.51 ± 2.68	−3.81 ± 0.20	−0.96	92.58	6.71	5.66

Estimates, mean ± SE, standard error; the statistical analysis is highly significant for all the experimental theses (*a*), for intercept and (*b*), slope; (*r*), correlation coefficient; (R^2^_adj_), R-squared adjusted for d.f. standard error; of (SEE), the estimate of the model; of (MAE), mean absolute error.

**Table 3 animals-09-00278-t003:** Multiple sample comparison of drying profiles (SD, VMD1, and VMD2) at different time intervals.

Drying Time (min)	SD	VMD1	VMD2	ANOVA
				*p*-Value
90	54.6 ± 0.12	56.4 ± 0.72	52.6 ± 3.91	0.218
120	45.9 ± 0.58	47.1 ± 0.29	43.9 ± 3.47	0.286
180	38.4 ± 0.57	36.4 ± 0.87	34.0 ± 3.52	0.195
240	36.4 ± 0.35 a	30.7 ± 1.19 b	28.6 ± 3.41 b	0.034
300	35.9 ± 0.24 a	28.1 ± 1.12 b	26.1 ± 3.16 b	0.009
360	35.7 ± 0.13 a	26.7 ± 0.89 b	24.9 ± 2.84 b	0.004

Means ± SD; values followed by different letters in the same row are significantly different (*p*-level < 0.05). ANOVA and Tukey post-hoc test were carried out (Statistica 8 software).

**Table 4 animals-09-00278-t004:** Microbial contaminations along the production chain of the insect meal for fish feed production (insects are grown on VDM in mass rearing conditions); mean values (Log cfu g^−1^) ± standard deviations of duplicate samples.

Products	Total Viable Aerobic Count	Enterobacteriaceae	*E. coli*	Enterococci	Sulfite-Reducing Clostridia Spores	Yeasts	Molds	Salmonella (in 25 g)
BSF rearing substrate (VMD)	6.55 ± 0.57	1.70 ± 0.71	<1.00	3.88 ± 1.64	2.15 ± 0.75	2.48 ± 0.76	<2.00	Absence
Dried BSF biomass	5.05 ± 0.06	<1.00	<1.00	<1.00	2.50 ± 0.14	2.30 ± 0.43	<2.00	Absence
Defatted BSF meal	4.35 ± 0.49	<1.00	<1.00	<1.00	1.54 ± 0.51	<2.00	<2.00	Absence
Experimental fish feed								
BSF 0 (Control)	4.23 ± 0.24	2.51 ± 0.63	<1.00	2.20 ± 0.17	<1.00	<2.00	<2.00	Absence
BSF 10 (10% BSF meal)	4.43 ± 0.38	2.76 ± 0.49	<1.00	1.30 ± 0.43	1.35 ± 0.49	2.15 ± 0.21	2.48 ± 0.67	Absence
BSF 20 (20% BSF meal)	4.25 ± 0.13	1.00 ± 0.00	<1.00	1.24 ± 0.34	1.15 ± 0.21	<2.00	<2.00	Absence
BSF 30 (30% BSF meal)	4.22 ± 0.45	<1.00	<1.00	<1.00	<1.00	<2.00	<2.00	Absence

**Table 5 animals-09-00278-t005:** Nutritional analyses of the BSF biomass produced on VMD in mass rearing condition.

% d.m.	Mean ± SD
Dry matter	97.00 ± 0.31
Crude protein	39.42 ± 0.16
Ash	7.08 ± 0.00
Fat	35.62 ± 0.27
Chitin	4.02 ± 0.02
Nitrogen-free extracts	13.86 ± 0.14
Starch	1.82 ± 0.05
Glucose	0.30 ± 0.00

**Table 6 animals-09-00278-t006:** Amino acid content of BSF biomass produced on VMD in mass rearing conditions (mg/g d.m).

Amino Acids	Mean ± SD	Amino Acids	Mean ± SD
Aspartate	37.89 ± 0.43	Cysteine	0.46 ± 0.02
Glutamic acid	64.87 ± 0.36	Valine	24.68 ± 0.57
Serine	17.94 ± 0.15	Methionine	17.62 ± 0.83
Histidine	12.36 ± 0.74	Phenylalanine	18.84 ± 0.06
Glycine	22.90 ± 0.01	Isoleucine	15.86 ± 0.09
Threonine	21.53 ± 0.69	Leucine	26.98 ± 0.13
Arginine	48.30 ± 1.04	Lysine	19.78 ± 0.31
Alanine	38.85 ± 0.11	Proline	14.56 ± 0.32
Tyrosine	20.46 ± 0.32	Tryptophan	4.27 ± 0.08

**Table 7 animals-09-00278-t007:** Growth performances of rainbow trout fed the experimental diets (n = 3 tanks).

Diet	BSF0	BSF10	BSF20	BSF30
IBW(g)	67.01 ± 1.71	66.38 ± 2.51	65.63 ± 0.42	66.95 ± 2.31
FBW(g)	223.20 ± 23.67	220.34 ± 29.60	216.97± 26.16	221.74 ± 22.25
WG(g)	156.86 ± 4.33	154.20 ± 6.04	146.89 ± 8.03	152.30 ± 10.18

Means ± SE; IBW, initial body weight; FBW, final body weight; WG, weight gain. Values do not differ significantly according to one-way ANOVA (IBM SPSS Statistics version 25).

**Table 8 animals-09-00278-t008:** Growth parameters of earthworms reared on BSF rearing leftovers, VMD, or simple peat moss (control).

	BSF Leftovers	VMD	Peat Moss
Initial earthworm number	10	10	10
Initial earthworm weight (g)	3.17 ± 0.32	2.90 ± 0.15	2.90 ± 0.62
Final earthworm number	21 ± 7	41 ± 21	8 ± 3
Final earthworm weight (g)	3.69 ± 1.12 a	3.10 ± 0.25 a	1.58 ± 1.73 b
Number increase	11 ± 7 a	31 ± 21 a	−1.33 ± 0.78 b
Weight increase (g)	0.52 ± 0.33 a	0.20 ± 0.33 a	−1.32 ± 1.38 b

Values are reported as mean ± SD. In the same row different letters denote significant differences (One-way ANOVA and Tukey’s test—Statsoft).

**Table 9 animals-09-00278-t009:** Chemical analyses of the three growing substrates and of resulting organic soil amendments.

Dry Matter (%)	Ash *	N * (TKN)	EE *	CF *	A *	Nitrates ^§^	N * (CNS)	C * (CNS)	C/N
Peat moss (63.74) *	3.05	0.99	0.96	32.33	0.022	39.74	1.22	51.92	42.56
VMD (19.78) *	5.18	1.18	2.67	30.63	0.056	N.D.	5.50	51.93	32.25
BSF leftovers (90.70) *	9.77	2.25	3.12	34.68	0.127	5.73	2.45	47.46	19.37
Peat moss + Ef (19.63) *	4.10	1.07	1.14	33.36	0.017	61.56	1.22	52.33	42.89
VMD + Peat moss + Ef (13.90) *	4.75	1.30	0.88	31.99	0.095	2035.26	1.50	51.34	34.23
BSF + Peat moss + Ef (18.36) *	7.07	1.84	0.85	30.55	0.228	8185.37	2.06	49.86	24.20

Ef, after *E. fetida* rearing; N, nitrogen; TKN, measured by Kjeldahl method; EE, ether extracts; CF, crude fiber; A, ammonia (ammoniacal nitrogen); C, carbon; CNS, measured by elemental analyzer; * % dry matter; ^§^ mg/kg dry matter.

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
