# Peer review of "A First Attempt to Produce Proteins from Insects by Means of a Circular Economy"

_animals, 2019, doi:10.3390/ani9050278_

Round 1

Reviewer 1 Report

Reviewer 1

An environmentally closed loop for food supply in which fruit and vegetable waste from markets became rearing substrate for Hermetia illucens (Hi) is compared to a standard diet for Diptera in terms of larval growth, waste reduction index, and overall substrate degradation and processing such as drying and oil extraction from Hi and use in aquafeed and and soil amendment.

The paper  suggests a pathway according to the principle of circular economy to optimize the use of organic wastes.  To this respect it considers different aspects and methodologies, giving a multidisciplinary approach as needed to the topic. It could be appropriate for Animals.

Anyway there are some critical points that should be taken into account and need to be seriously revised as  some information are missing and there several weaknesses and shortcomings.

Minor comments

Make the expression of data uniform within the text. Authors use mean values ± SD, mean and SD in a separate column or row,  and different letter, ANOVA P-value, or *** to indicate statistical significance.  

Detailed comments

Line 100 Use also the scientific name of earthworms as stated for the other previously cited species.

Line 113 The term genetic background do not seem appropriate to simply define “ Hi collected from the field (Lombardy, Northern Italy).” It doesn’ t seem to define any specific strain or genetic characteristic.

Line 117  Specify how the experimental diets were formulated, “appropriately minced” appears too generic.

Line 126 The tests to establish the best rearing conditions are not clear as in the previous paragraph only a SD vs a VDM diet was compared under fixed environmental conditions.

Line 163 It is supposed that the same, 600g) biomass of frozen BSF was employed for SD and VMD, but this is not clear.

Line 205 Experimental diet and fish feeding trial are both aspects poorly described in the paragraph. More detail should be given.

Line 211 Please specify the feeding level and detail the diet formula. Authors only say that HI meal replaced different level of fish meal, but nothing it is said on the dietary protein and lipid level, if the diet were isoproteic and which other ingredients were used to compensate the FM reduction. If the authors refer to the study of Terova et al., (2019), this is not available to the reviewer, yet. Which was the animal feeding ratio, if any…

Line 243 and 246 Peat moss

Paragraph 2.12 The order of the sentences is a bit confusing. It would be better to refrase the paragraph.

Line 274 “….although no significant variations between the two experimental theses were recorded” this conclusion seems to be in contrast with  “the length of the larval stage was significantly different in the two conditions and the average of maximum weight achieved before prepupal stage by VMD group was lower than that of SD group,…” please clarify unambigously.

Table 1 and 4 and 9 The same  letter in the row are not needed as no significant differences were detected

Line 332 In the Material and Method section 2.4 it is specified that three replicates from each condition were used in the analysis, anyway the differential expression level of midgut genes is expressed in Table 2 by a single figure. A mean and a variability should be indicated and a statistical analysis should be performed to verify statistical significance.

Table 2 Could the authors tempt some correlation between the gene expression values reported in Table 2 and the growth parameter/development time registered and shown in Table 1 for laboratory scale reared larvae on the SD and VMD substrate.   

Line 376 Crude ether extract

Line 408 “amino acidic” modify in amino acid

Line 409-410 “ it is indispensable to understand how to balance the insect meal nutrients with the remaining fish feed ration.” The sentence needs to be reformulated.

Line 412 The composition and microbiological analysis of the partially defatted BSF meal presented in Table 5 and Table 6 is referred to the one reared on VMD. It would be of help to remind this in the title of the Tables.  Do the authors observed any difference in larvae nutritional composition according to rearing substrate?

Table 7. Modify title in Amino acid content (mg/g d.m.) of BSF biomass reared on VMD. Anyway in line 421 authors say that “the amino acid profile (it is a content? ) in BSF pre-pupae, accordingly with the different rearing substrates”. Please avoid ambiguity on this aspect.  Specify the number of replicates. Data should be presented as mean ± SD in the same column. Align the value for Tyrosine

Line 450 Only fish growth was shown as no other performance parameter was considered. In growth performance studies it is important to know nutrient bio-availabilties, and this requires information about feed and nutrient digestibility as well as the efficiency with which nutrients are used for growth. The authors did not collect these data; this is a design weakness that limit the consideration on differences among the different Hi meal inclusion level. It is suggested to limit data to growth in the title  

Table 9. Modify rows with columns

Author Response

Reviewer 1
An environmentally closed loop for food supply in which fruit and vegetable waste from markets became rearing substrate for Hermetia illucens (Hi) is compared to a standard diet for Diptera in terms of larval growth, waste reduction index, and overall substrate degradation and processing such as drying and oil extraction from Hi and use in aquafeed and and soil amendment.
The paper suggests a pathway according to the principle of circular economy to optimize the use of organic wastes. To this respect it considers different aspects and methodologies, giving a multidisciplinary approach as needed to the topic. It could be appropriate for Animals.
Anyway there are some critical points that should be taken into account and need to be seriously revised as some information are missing and there several weaknesses and shortcomings.
Minor comments
Make the expression of data uniform within the text. Authors use mean values ± SD, mean and SD in a separate column or row, and different letter, ANOVA P-value, or *** to indicate statistical significance.
Expression of data was made uniform in all the Tables. It was not possible to substitute asterisks with letters in Figure 7 because of some specific constraints due to the used software.
Detailed comments
Line 100 Use also the scientific name of earthworms as stated for the other previously cited species.
It was done.
Line 113 The term genetic background do not seem appropriate to simply define “Hi collected from the field (Lombardy, Northern Italy).” It doesn’ t seem to define any specific strain or genetic characteristic.
The term was removed from the title.
Line 117 Specify how the experimental diets were formulated, “appropriately minced” appears too generic.
An explanation was added (line 141-143) with regard to VMD diet (the SD diet is defined according with the reported reference): “a diet composed of a mix of fruit and vegetables (zucchini, apple, potato, green beans, carrot, pepper, orange, celery, kiwi, plum, aubergines) cut into small pieces of about 5 mm (VMD)”.
Line 126 The tests to establish the best rearing conditions are not clear as in the previous paragraph only a SD vs a VDM diet was compared under fixed environmental conditions.
A sentence was added in both paragraph 2.1 (line 146-148) and 2.2 (160-161) explaining that: “A series of preliminary tests was carried out to establish the optimum quantity of feed”
Line 163 It is supposed that the same, 600g) biomass of frozen BSF was employed for SD and VMD, but this is not clear.
A sentence was added (line 186-189) to better define the used BSF biomass.
Line 205 Experimental diet and fish feeding trial are both aspects poorly described in the paragraph. More detail should be given.
Line 211 Please specify the feeding level and detail the diet formula. Authors only say that HI meal replaced different level of fish meal, but nothing it is said on the dietary protein and lipid level, if the diet were isoproteic and which other ingredients were used to compensate the FM reduction. If the authors refer to the study of Terova et al., (2019), this is not available to the reviewer, yet. Which was the animal feeding ratio, if any…
We agree with the reviewer and we have included in the revised version of this paragraph the requested information as follows:
Specifically, one control diet with 0% (BSF 0) and three experimental diets with 10% (BSF 10), 20% (BSF 20), and 30% (BSF 30) of BSF meal, were formulated to be isonitrogenous [crude protein: about 49 g/100 g dry matter], isolipidic [ether extract: about 18 g/100 g dry matter] and isoenergetic [gross energy about 22.90 MJ/kg d.m.]. In order to maintain diets isonitrogenous, isolipidic and isoenergetic, and due to the different chemical composition of BSF compared to FM, with the increase of BSF meal inclusion, soybean oil and wheat bran amounts were correspondingly modified.
The apparent digestibility coefficients of dry matter, proteins, ether extract and gross energy of each diet were assessed through an in vivo digestibility trial.
Feed was manually distributed and feeding rate was restricted to 1.5% of biomass for the entire duration of the experiment.
For other details on experimental diets and fish feeding please refer to Terova et al., 2019, which has now been published on Reviews in Fish Biology and Fisheries journal. We have updated this reference in the MS adding a link to the on line version, so the study is now available to the reviewer and to the readers (https://doi.org/10.1007/s11160-019-09558-y).
Line 243 and 246 Peat moss
It was corrected.
Paragraph 2.12 The order of the sentences is a bit confusing. It would be better to refrase the paragraph.
It was rephrased as follows: “The following software programs were used for statistical analyses: Statistica 8 (StatSoft), IBM SPSS Statistics version 25, GraphPad Prism5 (GraphPad Software, Inc., La Jolla, CA, USA). One-way ANOVA, followed by Tukey’s test, was carried out. Legends to figures and tables give details about statistical methods. Chemical, nutritional, and microbiological analyses were performed in duplicate”.
Moreover, some additional lines were added in the legends of the tables reported in the text to better clarify the used software on case by case basis.
Line 274 “….although no significant variations between the two experimental theses were recorded” this conclusion seems to be in contrast with “the length of the larval stage was significantly different
in the two conditions and the average of maximum weight achieved before prepupal stage by VMD group was lower than that of SD group,…” please clarify unambigously.
The sentence was modified as follows: “In fact, although the maximum weight achieved before prepupal stage was not significantly different in the two groups, the duration of the larval stage was longer in VMD group than in SD group (Figure 1)”.
Table 1 and 4 and 9 The same letter in the row are not needed as no significant differences were detected
It was modified according to the request.
Line 332 In the Material and Method section 2.4 it is specified that three replicates from each condition were used in the analysis, anyway the differential expression level of midgut genes is expressed in Table 2 by a single figure. A mean and a variability should be indicated and a statistical analysis should be performed to verify statistical significance.
According to the observations made by another referees the chapter on gene analysis and transcriptomics was removed from the paper.
Table 2 Could the authors tempt some correlation between the gene expression values reported in Table 2 and the growth parameter/development time registered and shown in Table 1 for laboratory scale reared larvae on the SD and VMD substrate.
According to the observation made by another referees the chapter on gene analysis and transcriptomics was removed from the paper.
Line 376 Crude ether extract
It was modified
Line 408 “amino acidic” modify in amino acid
It was modified
Line 409-410 “ it is indispensable to understand how to balance the insect meal nutrients with the remaining fish feed ration.” The sentence needs to be reformulated.
It was modified as follows: “in fact, any of the tested diets containing BSF meal should support fish growth”.
Line 412 The composition and microbiological analysis of the partially defatted BSF meal presented in Table 5 and Table 6 is referred to the one reared on VMD. It would be of help to remind this in the title of the Tables. Do the authors observed any difference in larvae nutritional composition according to rearing substrate?
It was added in the table titles. Yes, we observed a difference in the nutritional composition of BSF according to the rearing substrate. We did not add other tables to the Results section, as it is already quite complex; on the other hand, we included a sentence to explain this difference: “However, previous preliminary analyses (data not shown) highlighted a lower content of protein (5-6% less out
of the total d.m.) and a corresponding increase in the fat amount, in the BSF biomass obtained from larvae reared on VMD, in comparison to SD”
Table 7. Modify title in Amino acid content (mg/g d.m.) of BSF biomass reared on VMD. Anyway in line 421 authors say that “the amino acid profile (it is a content? ) in BSF pre-pupae, accordingly with the different rearing substrates”. Please avoid ambiguity on this aspect. Specify the number of replicates. Data should be presented as mean ± SD in the same column. Align the value for Tyrosine
It was modified as required. The number of replicates was 2 as specified in the paragraph about Statistical analysis (2.11). The data are now presented as required, the value for Tyrosine was aligned.
Line 450 Only fish growth was shown as no other performance parameter was considered. In growth performance studies it is important to know nutrient bio-availabilties, and this requires information about feed and nutrient digestibility as well as the efficiency with which nutrients are used for growth. The authors did not collect these data; this is a design weakness that limit the consideration on differences among the different Hi meal inclusion level. It is suggested to limit data to growth in the title
All the data mentioned by the reviewer has been collected from the feeding trial with rainbow trout, but much of the results are presented in Terova et al. 2019 [23]. So, only details vital to this manuscript are repeated in detail here whilst referencing Terova et al. [23] elsewhere.
However, by taking into account the comment of the reviewer, we have now included more fish growth performance and feed digestibility data, as requested by the reviewer. Accordingly, the paragraph “Fish growth performance” has been changed as follows:
“At the end of the feeding trial, all fish had tripled their initial body weight, and growth performance indexes such as weight gain (Table 7), and standard growth ratio (SGR) [23] were not affected by diet composition. Similarly, feed conversion rate (FCR) was comparable among the treatments and remained lower than one in all groups, meaning that all fish grew efficiently and including the H. illucens meal did not negatively affect diet palatability. For more details on fish growth parameters refer to Terova et al. [23]. With regard to the nutrient bioavailability, no differences were found among treatments for the apparent digestibility coefficient of dry matter, proteins, ether extract, and gross energy [23]. Crude protein digestibility was high and above 90% in all rainbow trout fed different experimental diets.”
Table 9. Modify rows with columns
It was done.

Reviewer 2 Report

Authors attempted to produce protein from black soldier fly larvae and use it as both a fish feed and earthworm meal.  This was in attempt to have a more circular economy and reduce waste.  

This manuscript was interesting and there were novel ideas here, however there is so much crammed into a single paper it's difficult to follow.  It would be worth breaking this into 2-3 manuscripts as too many details are missing from your materials and methods section that it makes the studies seem incomplete, and results bombard the reader with too much information.  

Specific comments:

The abbreviation for black soldier fly is confusing, as both BSF and Hi are used back and forth throughout, and not explained in enough detail as to if they are different.  Seems to be the same product?

There is no mention of, or introduction to the fact that microscopy, gene expression and microbial contamination analysis was to be done in this study at all until it's presented in the materials and methods.  How does this contribute to the circular economy?  This seems to be a study of its own.

There are significant details missing throughout the materials and methods section, which are needed in order to determine validity of experiments.  Additionally, materials and methods jumps around from fish diets, back to just analyzing black soldier fly larvae, making it difficult to follow.

Need significantly more information regarding the gene expression materials/methods and results.

There are so many results, and discussion jumps all over the place, making it difficult to follow and to really interpret important/novel findings.

With so many results presented, I was expecting the conclusions section to really tie all of the experiments together and make an overall conclusion of the studies main findings.  However, it falls short and is quite disappointing, just stating that it does not have the ambition of describing details for a large scale production?  How was that even the goal of the paper?  Leaves the reader quite confused.

Author Response

Authors attempted to produce protein from black soldier fly larvae and use it as both a fish feed and earthworm meal. This was in attempt to have a more circular economy and reduce waste.
This manuscript was interesting and there were novel ideas here, however there is so much crammed into a single paper it's difficult to follow. It would be worth breaking this into 2-3 manuscripts as too many details are missing from your materials and methods section that it makes the studies seem incomplete, and results bombard the reader with too much information.
The whole project, from which this paper arose, has already brought to writing four different manuscripts, on specific parts, which are quoted as references in this text. However, the idea of using unpublished data to delineate a possible new track in insect economy resulted in this text. According to your observations we deleted the part on genes and transcriptomics, which will be better described in a dedicated paper, and tried to better clarify the general context of our research.
Specific comments:
The abbreviation for black soldier fly is confusing, as both BSF and Hi are used back and forth throughout, and not explained in enough detail as to if they are different. Seems to be the same product?
It is the same. Hi was changed into BSF in the whole text. Furthermore, we explained better the difference between BSF larvae as described for the laboratory test and BSF biomass: “While under laboratory conditions BSF pre-pupae were easily isolated and discarded from the rearing substrate, under mass rearing conditions stage selection was not just as simple; therefore, when BSF biomass is mentioned in the text, it refers to a mix of mostly larvae and occasionally pre-pupae”.
There is no mention of, or introduction to the fact that microscopy, gene expression and microbial contamination analysis was to be done in this study at all until it's presented in the materials and methods. How does this contribute to the circular economy? This seems to be a study of its own.
We attempted to explain this part in the Abstract, lines 55-56, Introduction, lines 114-132, Conclusions, lines 796-813.
There are significant details missing throughout the materials and methods section, which are needed in order to determine validity of experiments. Additionally, materials and methods jumps around from fish diets, back to just analyzing black soldier fly larvae, making it difficult to follow.
According with the indications given by the other referees, we tried to clarify the adopted methodologies in Materials and methods. However, throughout the manuscript we refer to specific papers of us, dealing with the same matter, in which methodologies are described in full. To describe every part in detail would have been too long. We are aware that very different experiments were set up, but this is part of the novelty of the paper.
Need significantly more information regarding the gene expression materials/methods and results.
This part was deleted from the manuscript.
There are so many results, and discussion jumps all over the place, making it difficult to follow and to really interpret important/novel findings.
With so many results presented, I was expecting the conclusions section to really tie all of the experiments together and make an overall conclusion of the studies main findings. However, it falls short and is quite disappointing, just stating that it does not have the ambition of describing details for a large scale production? How was that even the goal of the paper? Leaves the reader quite confused.
Conclusions were re-written according to your observations, trying to clarify better the paper objectives and how we thought they were reached.

Reviewer 3 Report

This paper is well written and presents very interesting results. This study is a good example of interdisciplinary research and is very well executed. I recommend publishing. Minor corrections are required and some comments are added for the authors in the annotated file.

Author Response

Reviewer 3
It is important to asses the long-term use of this diet on insect growth. Deficiencies of some nutrients can accumulate across generations and epigenetic effects can also become detrimental after a few generations.
Yes, we agree with the referee’s observation and included it in the text (lines 621-628)
“Although BSF demonstrates the capacity to adapt to the food substrate to some extent, it is important to assess the long-term effect of the use of this diet on the insect growth. Deficiencies of some nutrients might accumulate across generations and epigenetic effects can also become detrimental after a few reproduction cycles. In this case, a possible solution to the problem might be to preserve a genetic stock for egg reproduction through rearing on SD and to use the VMD diet only to produce insect biomass for commercial uses”
At this temperature proteins are denatured and enzymes deactivated. Some coking occurs and some nutrients may be lost.
Yes, we agree with the referee’s observation and included it in the text (line 651-653).
This temperature represents a compromise between an unavoidable partial loss of nutritional function by some proteins and enzymes, which might be denatured, and the need to reduce the time required by the drying process

Round 2

Reviewer 2 Report

Much improved!  The added sections made things much more clear and especially with the obvious mention of other published studies from your lab that have more mythology details.  Additionally, the removal of the gene expression data was ideal, as that can be a publication in itself.  

It was noted that your conclusions were updated, but it would still be helpful to readers to have a short paragraph as to what your final thoughts are on the overall conclusions of the experiments.  It seems as if the goals of the paper are just repeated rather than what the authors think the outcome of the BSF meal is, and future applications in both fish diets and as earthworm substrates.  Is this a plausible product as a fish meal replacement?